

# Measuring QCD splittings with invertible networks

**Sebastian Bieringer[1], Anja Butter[1], Theo Heimel[1]\*, Stefan Höche[2],
Ullrich Köthe[3], Tilman Plehn[1] and Stefan T. Radev[4]**

**1** Institut für Theoretische Physik, Universität Heidelberg, Germany
**2** Fermi National Accelerator Laboratory, Batavia, IL, USA
**3** Heidelberg Collaboratory for Image Processing, Universität Heidelberg, Germany
**4** Psychologisches Institut, Universität Heidelberg, Germany

\* heimel@thphys.uni-heidelberg.de

## Abstract

QCD splittings are among the most fundamental theory concepts at the LHC. We show how they can be studied systematically with the help of invertible neural networks. These networks work with sub-jet information to extract fundamental parameters from jet samples. Our approach expands the LEP measurements of QCD Casimirs to a systematic test of QCD properties based on low-level jet observables. Starting with an toy example we study the effect of the full shower, hadronization, and detector effects in detail.

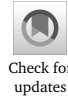

## Content

## 1 Introduction

The upcoming Run 3 and HL-LHC are starting an era of precision physics at hadron colliders. With this perspective we need to re-think our strategies for measurements, interpretation

frameworks, and first-principle theory predictions. A big step in the direction of Machine Learning (ML) based measurements has been made in jet classification based on low-level detector output. It starts from the observation that subjet taggers benefit immensely from multivariate approaches [1, 2], combined with the ability of modern convolutional networks to extract subjet information [3–8]. Alternatively, we can feed a network the subjet 4-momenta [9–12], or change the architecture to recurrent networks [13] or point clouds [14, 15]. Top tagging is an especially interesting subjet problem, because the tagger output is theoretically well defined and the training can be done on data only. This makes top taggers an excellent conceptual testing ground [16], including the crucial question how to control uncertainties [17, 18]. Already in the context of taggers, the focus on data-driven ML-applications becomes a major problem for particle physics when it breaks our central promise of understanding LHC collisions entirely in terms of fundamental physics. The question then becomes how we can use these new approaches to improve our understanding of QCD [19, 20].

From a theory perspective, the LHC objects described by the simplest fundamental laws, at least to leading order, are parton showers [21]. In perturbative QCD their entire behavior can be described by the quark-gluon interaction and the triple gluon interaction. Their leading kinematic behavior can be understood as logarithmically enhanced collinear and soft splittings, in which we can replace the particle interactions by a set of simple splitting kernels. Predictions beyond this simple approximation are an active research field in view of the coming LHC runs [22–27]. This progress motivates the question what kind of fundamental QCD properties we can test in terms of parton splittings defining relatively simple physics objects.

There exists a history of extracting QCD properties from collider data. Before the LHC era, a global strategy combining LEP-measurements like jet and event shapes, scaling violation, fragmentation functions, $Z$-pole measurements, and $\tau$-decays with low-energy $e^+e^-$-data extracted the value of the strong coupling as $\alpha_s(m_Z) = 0.1211 \pm 0.0021$ [28, 29]. Obviously, this measurement has since been improved by hadron collider and HERA data. In addition, LEP data has been used for another fundamental QCD measurement, namely QCD color factors or specifically quadratic $SU(3)$ Casimir invariants as they appear in QCD splittings. They can be extracted from a variety of jet or event shapes in 3-jet and 4-jet final states. To start with, measurements in 3-jet events by OPAL give $C_A/C_F = 2.232 \pm 0.14$ [30], while the 3-jet measurements by DELPHI lead to $C_A/C_F = 2.26 \pm 0.16$ [31]. Studies of the electroweak 4-jet kinematic by ALEPH gives $C_A = 2.93 \pm 0.60$ and $C_F = 1.35 \pm 0.27$ [32], a similar analysis by OPAL quotes $C_A = 3.02 \pm 0.56$ and $C_F = 1.34 \pm 0.30$ [33]. In both 4-jet analyses there exists a strong, positive correlation between $C_A$ and $C_F$. Event shapes [34–39], similar to modern jet shapes [40], can be used to extract the same parameters and give $C_A = 2.84 \pm 0.24$ and $C_F = 1.29 \pm 0.18$ [41]. The combined analysis reports [28, 29]

$$C_A = 2.89 \pm 0.21 \quad \text{and} \quad C_F = 1.30 \pm 0.09 \,, \tag{1}$$

with both measurements being clearly systematics limited.

In this paper we propose a way to adapt these measurements for the LHC era. An obvious path would be a comprehensive analysis of multi-jet production, which would probably have to be combined with the global extraction of parton densities together with the strong coupling constant. Instead of such a comprehensive global analysis we base our study on parton shower data, which does not require us to understand parton densities or mass effects or large electroweak corrections. Our goal is to put the above LEP measurements into a context of QCD measurements at the LHC and to develop a framework for learning the properties of QCD splittings. At the subjet level there exists a range of observables for which we can compare precision predictions with precision measurements [42–44]. On the other hand, parton showers are a prime example for precision simulations, so we will follow the orthogonal approach of extracting fundamental QCD parameters using simulation-based inference. The

inspiration for our analysis are new methods referred to as likelihood-free inference at the event level [45, 46]. In both cases, the crucial ingredient is first-principle precision simulations with full control over the underlying hypothesis and over its theoretical self-consistency in describing the corresponding objects.

Technically, our BayesFlow approach [47] is based on the conditional version [48, 49] of invertible networks (INNs) [50–52], a specific realization of normalizing flows [53–56]. These networks have been studied in relation to phase space generation [57–60], event generation [61], anomaly detection [62], detector and parton shower unfolding [63], and density estimation [64]. We will introduce our QCD inference framework in Sec. 2 and illustrate our splitting kernel measurements in Sec. 3. Finally, we will attempt a more realistic benchmarking for the SHERPA [65] shower with hadronization and detector effects in Sec. 4.

## 2 INN-Inference and BayesFlow

**INN** The workhorse of our inference method is an invertible neural network (INN) which realizes a normalizing flow [55] between model parameters $m$ viewed as random vectors and a latent random vector $z$. Such an INN with the trainable parameters $\theta$ represents an easily invertible function $g_\theta(m)$ which transforms $m$ into $z$, whereas its inverse $\bar{g}_\theta(z)$ transforms $z$ back into $m$. This way the INN simultaneously encodes both directions of a bijective mapping between $m$ and $z$ via a single set of parameter $\theta$ learned through gradient-based optimization.

Coupling flows are a widely used invertible architecture, since they are capable of learning highly expressive transformations with tractable Jacobian determinants [52, 55]. We construct our INNs by composing multiple affine coupling layers [51, 66] into a composite invertible architecture. A single coupling layer $g_{\theta_j}$ splits its input vector $m$ into two halves, $m = (m^A, m^B)$, to obtain $z = (z^A, z^B)$ via the bijective transformation

$$\begin{pmatrix} z^A \\ z^B \end{pmatrix} = \begin{pmatrix} m^A \odot e^{s_2(m^B)} + t_2(m^B) \\ m^B \odot e^{s_1(z^A)} + t_1(z^A) \end{pmatrix} \quad \Leftrightarrow \quad \begin{pmatrix} m^A \\ m^B \end{pmatrix} = \begin{pmatrix} (z^A - t_2(m^B)) \odot e^{-s_2(m^B)} \\ (z^B - t_1(z^A)) \odot e^{-s_1(z^A)} \end{pmatrix}. \tag{2}$$

By construction, this bijection works independently of the form of the functions $s$ and $t$. For our application, $s$ and $t$ are realized via feed-forward neural networks with trainable parameters $\theta_j$ in each coupling layer. The Jacobian of each coupling flow layer is the product of two triangular matrices

$$\frac{\partial g_{\theta_j}(m)}{\partial m} = \begin{pmatrix} \mathbb{1} & 0 \\ \text{finite} & \text{diag } e^{s_1(z^A)} \end{pmatrix} \begin{pmatrix} \text{diag } e^{s_2(m^B)} & \text{finite} \\ 0 & \mathbb{1} \end{pmatrix}, \tag{3}$$

making its determinant fast to compute. Much effort has gone into improving the efficiency of invertible coupling layers. We use the all-in-one coupling layer with three additional features [51, 52]. First, each layer incorporates a fixed permutation before splitting its input, to ensure that each component in the final $z$ is influenced by each component of the initial $m$. Second, it includes a global affine transformation to induce a bias and linear scaling. Third, it applies a bijective soft clamping after the exponential function in Eq.(2) to prevent instabilities from divergent outputs [48].

We combine multiple coupling layers to increase the expressiveness of the learned transformation. This is possible because a combination of invertible functions is again invertible and its Jacobian is the product of the individual Jacobians. For $J$ coupling layers, our composite INN is given by

$$z = g_{\theta_J} \circ g_{\theta_{J-1}} \circ \cdots \circ g_{\theta_1}(m), \tag{4}$$

with trainable parameters $\theta = (\theta_1, \ldots, \theta_J)$ and the inverse

$$m = \bar{g}_{\theta_1} \circ \cdots \circ \bar{g}_{\theta_{J-1}} \circ \bar{g}_{\theta_J}(z). \tag{5}$$

Such a composition can be viewed as transforming or normalizing a complicated, intractable source distribution $P(m)$ into a much simpler, tractable, $P(z)$ prescribed by the optimization criterion.

**Conditional INN**  To recover model parameters from a set of measurements $x$ we need to augment the INN architecture in two ways. First, we turn the invertible network into a conditional invertible network (cINN). A cINN still defines a bijective mapping between $m$ and $z$, but the functions $s$ and $t$ in each coupling layer take a set of measurements as an additional input, $z = g_\theta(m; x)$. Second, since the number of measurements can vary in practice, we introduce a relatively small summary network $h_\psi$ with trainable parameters $\psi$. It reduces measurements of variable size to fixed-size vectors, $\tilde{x} = h_\psi(x)$, by respecting the probabilistic symmetry of the measurements [47]. For independent measurements we use a permutation invariant summary network such that its output is invariant under the ordering in $x$ [47]. The summary network does not have to be invertible, since its output is concatenated with $m$ and fed to $s$ and $t$, but not directly mapped to $z$. Moreover, the two networks can be trained together to approximate the true parameter posterior $P(m|x)$ via an approximate posterior $Q$ defined by the network weights. Due to the change of variable formula, this approximate posterior is given by

$$Q(m|x) = P(z) \left| \det\left( \frac{\partial z}{\partial m} \right) \right| \quad \text{with} \quad z = g_\theta(m; h_\psi(x)), \tag{6}$$

and it represents the probabilistic solution to the inverse inference problem.

Together, the cINN and the summary network minimize the expected Kullback-Leibler divergence between the true and approximate posterior. Ignoring all terms that do not depend on the network parameters, this corresponds to minimizing the expected negative logarithm of the approximate posterior,

$$\min_{\theta, \psi} \langle \mathbb{KL}\left( P(m|x) \,||\, Q(m|x) \right) \rangle_{m,x} \sim \min_{\theta, \psi} \langle -\log Q(m|x) \rangle_{m,x} + \text{const.} \tag{7}$$

Finally, we can apply a coordinate transformation for the bijective mapping and enforce a Gaussian noise distribution with mean zero and width one for the latent distribution $P(z)$, so the loss function becomes

$$
\begin{aligned}
L(\theta, \psi) &= -\left\langle \log P(g_\theta(m; h_\psi(x))) + \log \left| \frac{\partial g_\theta(m; h_\psi(x))}{\partial m} \right| \right\rangle_{m,x} \\
&= -\left\langle -\frac{1}{2} \left\| g_\theta(m; h_\psi(x)) \right\|^2 + \log \left| \frac{\partial g_\theta(m; h_\psi(x))}{\partial m} \right| \right\rangle_{m,x}.
\end{aligned} \tag{8}
$$

This loss guarantees that the networks recover the true posterior under perfect convergence [47].

**Inference**  BayesFlow [47] provides a cINN framework which we can use to measure fundamental QCD parameters. From the inversion of a detector simulation and QCD radiation [63] we know how, given a single detector-level event, the cINN generates samples from a probability distribution over the phase space of the hard scattering. For the jet inference presented in this paper, the BayesFlow setup corresponds to this unfolding setup, in which we replace the parton-level phase space with the model parameter space and the detector-level phase space

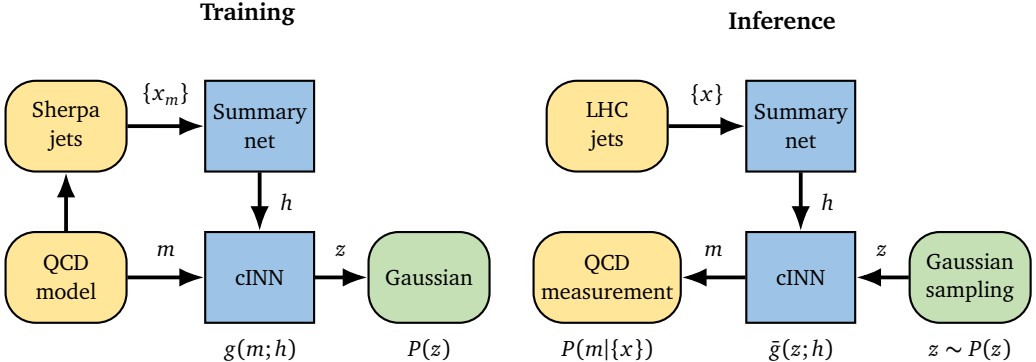

Figure 1: BayesFlow setup of the cINN for training and inference [47].

with the simulated data. In Fig 1 we give a graphical illustration of the inference setup, for the training and the inference phases.

To train the BayesFlow networks we use the fact that we can simulate an arbitrary number of jets fast. This allows us to employ mini-batch gradient descent to approximate the expectation in the above optimization criterion via its Monte-Carlo empirical mean. Moreover, if we train the networks on jet samples of varying size, we can use them on data samples with any size, as long as this size is within the domain of the pre-defined distribution over sample sizes. The networks will approximate the correct push-forward from a given prior $P(m)$ in model space to a posterior $P(m|x)$ contingent on a set of measurements $x$. When the test sample size leaves the training domain the posterior accuracy will degrade. In case we need to analyse larger data sets we can then follow the Bayesian logic behind the BayesFlow framework [47] and use the posterior from an earlier measurement as a prior.

## 3  Idealized jet measurements

Before applying BayesFlow to LHC jets including hadronization and detector simulation, we define our theory assumptions and test the corresponding model on an idealized data set using a toy shower [67]. That will give us an idea what kind of measurement we could aim for and will also allow for some simple benchmarking. We have checked that this toy shower agrees with the full SHERPA shower, except that we do not include the effects from the 2-loop cusp anomalous dimension.

**Theory setup**  The physics goal in our paper is to understand the QCD splittings building up parton showers. In the leading collinear approximation these kernels relate the amplitudes of an $n$-particle hard process $\mathcal{M}_n$ to the amplitude with an additional parton $\mathcal{M}_{n+1}$ [68]

$$\overline{|\mathcal{M}_{n+1}|^2} \simeq \frac{2g_s^2}{p_a^2} \hat{P}(z,y) \overline{|\mathcal{M}_n|^2} \,, \tag{9}$$

where $g_s$ is the strong coupling, $p_a^2 = (p+k)^2$ the invariant mass of the splitting parton, and $\hat{P}(z,y)$ the un-regularized splitting kernel. It depends on the energy fraction $z$ and the momentum transfer $y$, which in combination with a Catani-Seymour spectator momentum $p_s$ can be combined to the transverse momentum in the splitting,

$$z = \frac{pp_s}{pp_s + kp_s}$$
$$y = \frac{pk}{pk + pp_s + kp_s} \quad \Rightarrow \quad yz(1-z) \propto p_T^2 \,. \tag{10}$$

In massless QCD some of the kernels $\hat{P}$ include infrared divergences. They can be partially fractioned to remove soft double counting, giving us the three QCD splittings [69]

$$P_{qq}(z,y) = C_F\left[D_{qq}\frac{2z(1-y)}{1-z(1-y)} + F_{qq}(1-z) + C_{qq}yz(1-z)\right],$$

$$P_{gg}(z,y) = 2C_A\left[D_{gg}\left(\frac{z(1-y)}{1-z(1-y)} + \frac{(1-z)(1-y)}{1-(1-z)(1-y)}\right) + F_{gg}z(1-z) + C_{gg}yz(1-z)\right],$$

$$P_{gq}(z,y) = T_R\left[F_{qq}\left(z^2 + (1-z)^2\right) + C_{gq}yz(1-z)\right].$$

(11)

In this form we include a set of parameters which to leading order in perturbative QCD are given by

$$D_{qq,gg} = 1 \qquad F_{qq,gg} = 1 \qquad C_{qq,gg,gq} = 0. \tag{12}$$

The splitting kernels given in Eq.(11) define the fundamental physics hypothesis of our measurements, which should generalize the $C_A/C_F$ studies from LEP [28,29]. This hypothesis is flexible enough to accomodate precision predictions consistently with the kinematics of parton shower data at the LHC. Concerning its uniqueness, in standard parton showers, $D$ is typically modified to include a universal $K$-factor that coincides with the two-loop cusp anomalous dimension and resums sub-leading logarithms arising from the collinear splitting of soft gluons [70]. For simplicity, we will set these terms to zero in our toy shower. Within SHERPA, they are included through a modified running coupling. The second term reflects the leading terms in $p_T$, in our case truncated in the strong coupling. The rest terms $C_{ij}$ are, in our case, defined by the appearance of $p_T^2$. Generally, we only consider Eq.(11) as a first attempt for an appropriate theory hypothesis, which might have to be slightly modified according to the precision simulation framework used for the actual analysis. Another motivations for a modified theory hypothesis could be specific parametrizations to, for instance, incorporate quantum effects or $1 \to 3$ splittings. We skip this option because we will see that already the global rest terms of Eq.(11) challenge our simulated data. As alluded to in the Introduction, a caveat concerning the pre-defined theory hypothesis is common to all simulation-based or likelihood-free analyses.

We will vary the parameters in Eq.(12) away from the leading order QCD prediction, always making sure that the splitting kernels give positive splitting probabilities all over the collinear phase space by setting negative kernel values to zero. Given that the numerically leading contribution comes from the regularized pole, we can approximately identify the measurement of $D_{qq}$ and $D_{gg}$ with measurements of $C_F$ and $C_A$, as quoted in Eq.(1).

Table 1: Network setup and hyperparameters.

| | Symbol | Value |
|---|---|---|
| Number of parameters | $L$ | $2, 3$ |
| Maximum number of constituents | $F$ | $13$ |
| Jets per parameter point (variable/fixed) | $M$ | $10^2 \dots 10^5 / 10^4$ |
| Batch size | $N$ | $16$ |
| Batches per epoch | $E$ | $6250$ |
| Output dimension summary network | $S$ | $32$ |
| Fully connected summary net architecture | $S_i$ | $64,64,64,64,32,32$ |
| Coupling layers | $n_{\text{layers}}$ | $5$ |
| Fully connected coupling layer architecture | $s_i/t_i$ | $64,64,64$ |
| Epochs | $e$ | $10 \dots 40$ |
| Decay steps (toy shower/PF flow) | $n_s$ | $200 \dots 500 / 500 \dots 1000$ |
| Learning rate after $t$ batches | $\eta_t$ | $10^{-3} \cdot 0.99^{\lfloor t/n_s \rfloor}$ |
| Training/testing points | | $100\text{k} / 10\text{k}$ |

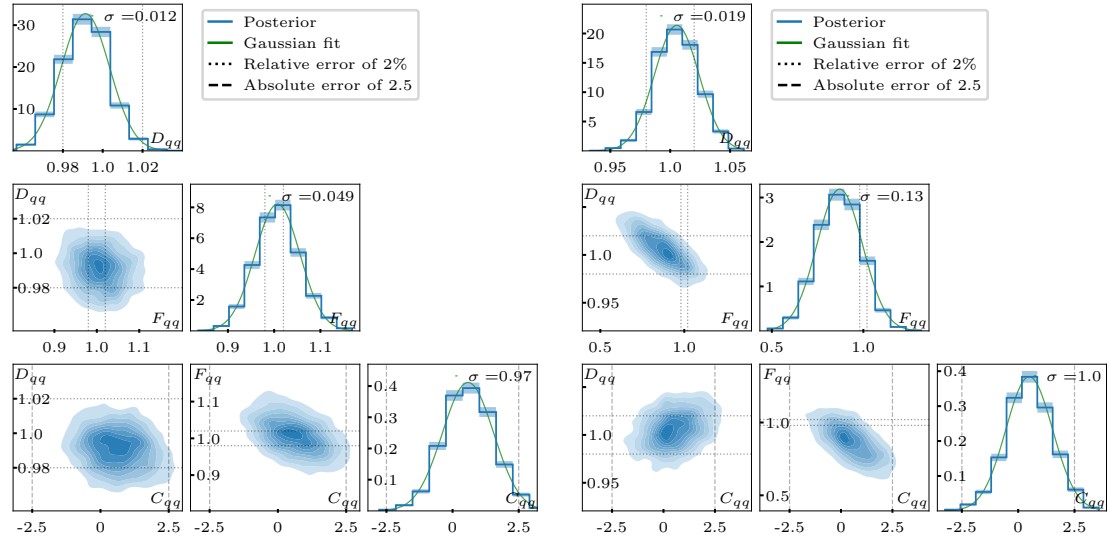

Figure 2: Posterior probabilities for the toy shower, gluon radiation only, $\{D_{qq}, F_{qq}, C_{qq}\}$. We assume SM-like jets and show results for truth-sorting (left) and for $k_T$-sorting (right).

**Data and network** To understand the proposed measurement in a controlled setup we simulate the on-shell process

$$e^+ e^- \to Z \to q\bar{q}, \tag{13}$$

assuming massless quarks and combined with a fast approximate parton shower cutoff at 1 GeV. Its phase space is completely defined by the scattering angle. For each event we apply the parton shower to one of the outgoing quarks, such that the second quark acts as the spectator for the the first splitting and we only consider one jet. For our jets sample we generally have

$$p_{T,j} < \frac{m_Z}{2}, \tag{14}$$

with the majority of jets at the upper boundary. After that, any other parton can act as the spectator. For this simple setup a jet reconstruction is not necessary, since we only simulate a single shower, and we neither include hadronization nor detector effects.

The network then analyses the set of outgoing momenta except for the initial spectator momentum. The list of constituents includes up to $F$ entries, and is zero-padded or cropped. For our training data we scan the parameter space $\{D_{ij}, F_{ij}, C_{ij}\}$ with $L = 2$ and 3 dimensions. For each parameter point we generate $M$ probabilistic showers. To observe the correct posterior contraction with the size of the test sample we train the network with variable $M$. During the training we use batches of size $N$. The input to the summary network per batch are $N \times M \times F$ 4-vectors. The output of the summary networks is mean-pooled over $M$ and has dimension $S$ for each batch, plus the value of $\sqrt{M}$, so $(S + 1)$ entries per batch, if the posterior contraction is trained.

The distribution of the number of jets $M$ over the $N$ batches can be adapted to the problem. We find that distributing the batches with $1/M$ is effective to counter the computational effort at high $M$. We will explicitly show that we retain enough high-$M$ information to guarantee the correct scaling of the error.

The cINN then provides a bijective mapping of the $L$-dimensional parameter space to the latent space of the same dimension, again per batch. The latent space is forced into a Gaussian

noise form, so we can sample from it to compute the probability distribution for a given set of $M_{eval}$ showers in model space. Values $M_{eval}$ not included in the training will lead to unstable results, if $\sqrt{M}$ was added to the summary network output. All parameters of the network architecture and the hyperparameters are given in Tab. 1. For the cINN we combine five coupling layers. The internal networks of the coupling layers, $s_{1/2}$ and $t_{1/2}$, are three fully connected layers with ELU activation. The summary network is built out of six fully connected layers with ReLU activation, ELU activation in the last layer, followed by average pooling. We use the Adam optimizer [71] with an exponentially decaying learning rate.

**Sorting**    Even though this constitutes an information backdoor, we first study what the network can extract if we order the constituents following their appearance in the shower. We refer to this unrealistic ordering of the constituents as truth-sorting. This means that after a splitting the daughter constituents are either appended to the end of the list or replace the mother momentum. Which daughter momentum overwrites the mother momentum is chosen by the showering algorithm. To avoid this backdoor we construct a similar ordering from the shower history given by a $k_T$-algorithm [72, 73], referred to as $k_T$-sorting. Since we simulate only a single jet, no jet radius has to be specified. We start with the first splitting and follow the hard constituents as the particles with the highest energy fraction in each splitting to determine the first entry of the list. The next entry is generated by following the hard constituents originating from the softer constituent of the earliest splitting. This is done for every splitting going from first to last. If an appearing parton is already assigned to the list, it is not assigned again.

**Gluon-radiation shower**    For our first test we restrict the shower to the $P_{qq}$ kernel of Eq.(11), implying that a hard quark successively radiates collinear and soft gluons. This way our 3-dimensional model space is given by

$$\{D_{qq}, F_{qq}, C_{qq}\} \,. \tag{15}$$

For the prior in model space we start with a uniform distribution over $[0.5, 2] \times [0, 4] \times [-10, 10]$ and train the network for 100000 randomly distributed points in model space.

In Fig. 2 we show the distribution of the posterior probabilities for $10^2 ... 10^5$ training jets per parameter point and 10000 test jets assuming SM-values. The fit confirms that all 1-dimensional posteriors are approximately Gaussian. Correlations among them are weak. As expected, we are more sensitive for the truth-sorting with its information backdoor. The reduced performance with the $k_T$-sorting indicates that pre-processing of the data plays an important role, and that information from jet algorithms should help. For the $k_T$-sorting the best-measured parameter in our toy model is the regularized divergence with a Gaussian standard deviation $\sigma(D_{qq}) = 0.019$. The finite terms are slightly harder to extract with $\sigma(F_{qq}) = 0.13$. Finally, the rest term with its assumed $p_T$-suppression comes with an even larger error, $\sigma(C_{qq}) = 1.0$. These increasing errors reflect the hierarchical structure of the splitting kernel. In addition to the reduced performance we also see that the information lost between truth-sorting and $k_T$-sorting induces visible correlations between the extracted model parameters. This correlation explains some of the loss in performance for instance in the $D_{qq}$ vs $F_{qq}$ plane, where the widths of the 1-dimensional posterior distributions are driven by the integration over the other parameters.

In Fig. 3 we show how the errors on these three model parameter change with the statistics of the test data set. Given the size of the training samples, $M = 10^2 ... 10^5$, we evaluate $M_{eval} = 10^3 ... 10^5$ SM-like jets and find that the Gaussian errors $\sigma(D_{qq})$, $\sigma(F_{qq})$, and $\sigma(C_{qq})$ all

---

All numerical results in this paper are also collected in Tab. 2.

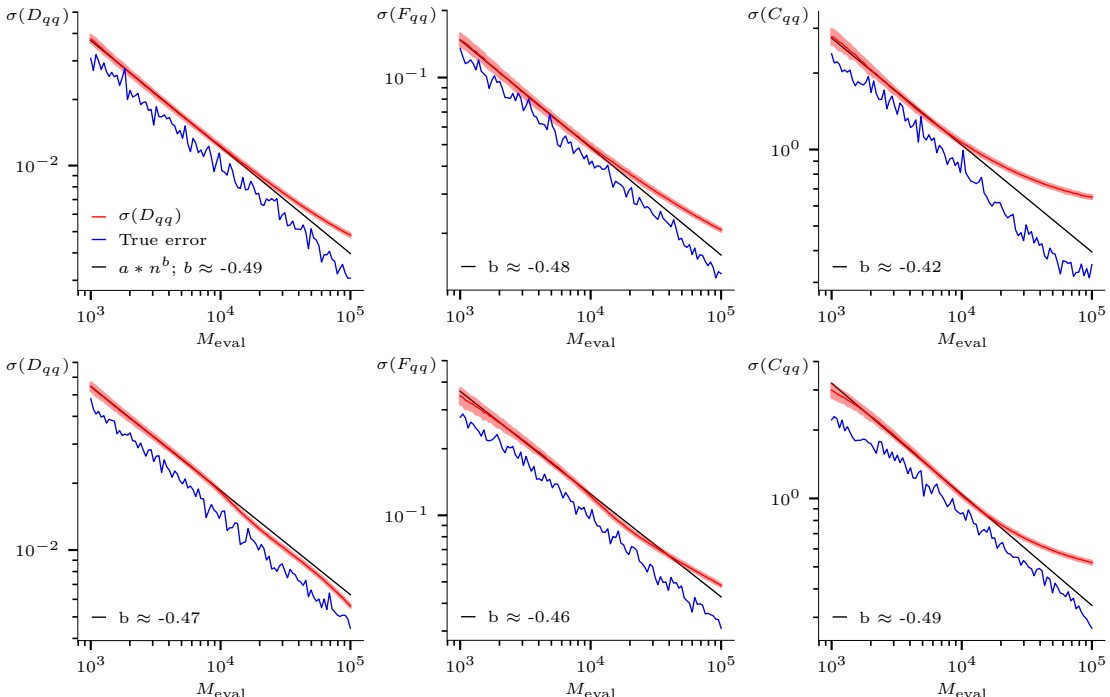

Figure 3: Uncertainty on $\{D_{qq}, F_{qq}, C_{qq}\}$ for gluon radiation only, as a function of the number of test jets. We show the standard deviation of the posterior (red) and the the absolute difference between the estimated and true parameters (blue) for truth-sorting (upper) and for $k_T$-sorting (lower). The black line is a fit to the posterior.

scale like $1/\sqrt{M_{\text{eval}}}$. This is expected for a statistically limited measurement. We also check the consistency of the network by comparing the reported standard deviation with the deviation between the central estimates and the truth. Altogether, the network performs exactly as expected, with the exception of a slight degradation in the challenging rest term $C_{qq}$ towards large test statistics.

**QCD splittings** In a second step, we include all three QCD splitting kernels from Eq.(11) and extract the soft-collinear divergences,

$$\{D_{qq}, D_{gg}\} . \tag{16}$$

Assuming that the leading logarithms really dominate the splittings and the sub-jet features, this measurement corresponds to measuring the two Casimirs $C_F$ and $C_A$. Because we only consider quark-jets from $Z$-decays, we expect the measurement of $D_{qq}$ or $C_F$ to be better, which is also what we observe in Fig. 4. Notably, for truth-sorting and for $k_T$-sorting there is no correlation between the two measurements, unlike for the standard LEP results.

For the final test on our toy model we include all QCD splitting kernels from Eq.(11) and determine the three $p_T$-suppressed rest terms

$$\{C_{qq}, C_{qg}, C_{gg}\} . \tag{17}$$

Hypothesis-wise this means that we assume that our predictions for the two leading contributions hold, and we want to estimate the size of an unknown contribution at higher power in $p_T$. The network is the same as for the gluon-radiation shower, with the results shown in Fig. 5. For $C_{qq}$ we first see that in the absence of the dominant contribution, the error drops slightly with respect to Fig. 2. The reason is that it is challenging for the network to disentangle the hierarchical structure of $\{D_{qq}, F_{qq}, C_{qq}\}$ in Fig. 2. For the other two rest terms, $C_{gg}$

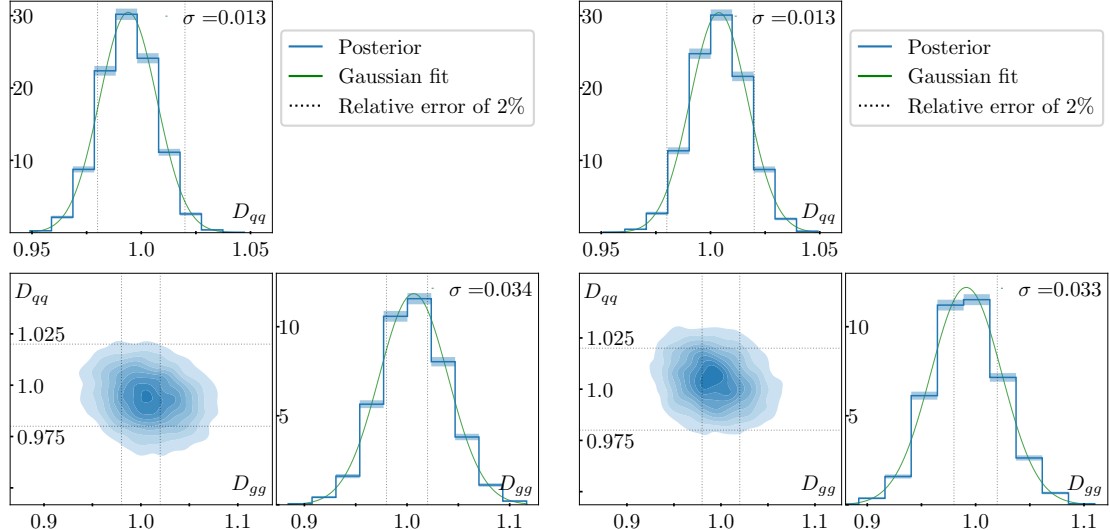

Figure 4: Posterior probabilities for the toy shower, soft-collinear leading terms for all QCD splittings, $\{D_{qq}, D_{gg}\}$. We assume SM-like jets and show results for truth-sorting (left) and for $k_T$-sorting (right).

and $C_{gq}$, we find significantly larger 1-dimensional errors and, related to these a strong anti-correlation. This correlation already exists for the truth-sorting case, so we expect it to remain in any realistic measurement.

**High-level observables**  To judge the impact of the low-level network input we can use the same setup as before, but feed high-level observables into the summary network. We use a set of six such observables [74], not all of them infrared and collinear safe. The simplest high-level observable for subjet analysis tracks the size of the splitting probabilities in terms of particle multiplicity ($n_{\mathrm{PF}}$) [75]. The width of the distributed radiation or girth is denoted at $w_{\mathrm{PF}}$ [76]. The effect of the soft divergence can be measured using $p_T D$ [77]. In addition, the two-point energy correlator $C_{0.2}$ is designed to separate quarks and gluons with an optimized power of $\Delta R_{ij}$ [78]. This defines a set of four standard observables

$$
n_{\mathrm{PF}} = \sum_i 1\,, \qquad\qquad w_{\mathrm{PF}} = \frac{\sum_i p_{T,i} \Delta R_{i,\mathrm{jet}}}{\sum_i p_{T,i}}\,,
$$
$$
p_T D = \frac{\sqrt{\sum_i p_{T,i}^2}}{\sum_i p_{T,i}}\,, \qquad\qquad C_{0.2} = \frac{\sum_{ij} E_{T,i} E_{T,j} (\Delta R_{ij})^{0.2}}{\sum_i E_{T,i}^2}\,. \qquad (18)
$$

In addition, we evaluate the highest fraction of $p_{T,\mathrm{jet}}$ contained in a single jet constituent and the minimum number of constituents which contain 95% of $p_{T,\mathrm{jet}}$ [79],

$$
x_{\max} \qquad \text{and} \qquad N_{95}\,. \qquad (19)
$$

The latter is obviously correlated with the number of constituents $n_{\mathrm{PF}}$.

In the left panels of Fig. 6 we compare the posteriors from the low-level observables and the six high-level observables for the gluon-radiation shower. The low-level results correspond to Fig. 2, and we remind ourselves that the truth-sorting with its information backdoor clearly leads to the best results. On the other hand, the $k_T$-sorting still delivers much better results than the high-level observables. In particular, the additional information from the complete

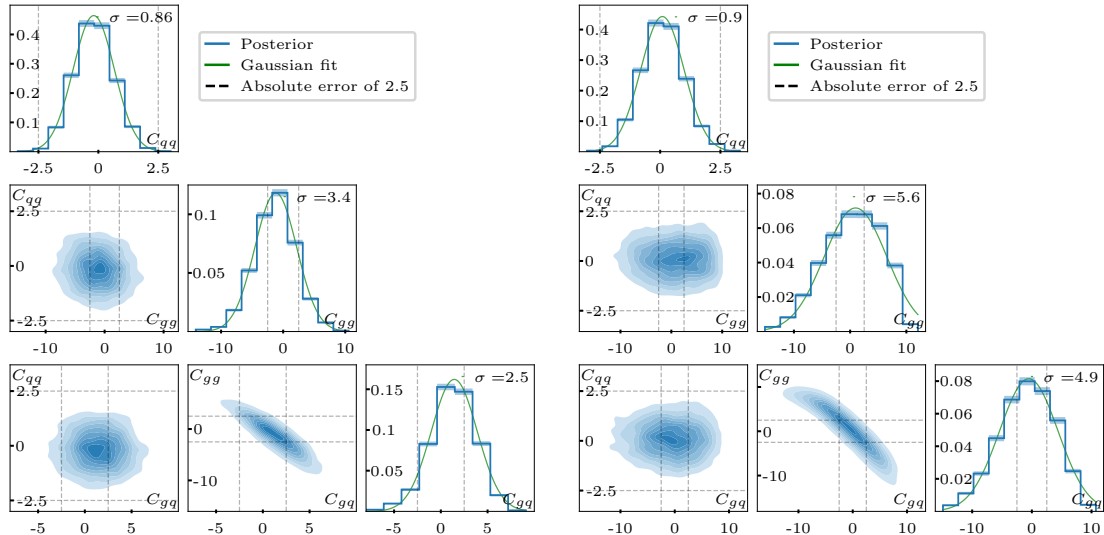

Figure 5: Posterior probabilities for the toy shower, $p_T$-suppressed rest terms for all QCD splittings, $\{C_{qq}, C_{gg}, C_{gq}\}$. We assume SM-like jets and show results for truth-sorting (left) and for $k_T$-sorting (right).

low-level information passed through the summary net reduces the correlations between the three measured parameters.

The right panels of Fig. 6 show the same $p_T$-suppressed rest terms as we see in Fig. 5, but including the projected measurements from the high-level observables. Again, the truth-sorting should not be taken as a realistic benchmark, but even the $k_T$-sorting avoids the non-Gaussian structures we see for the high-level observables. Aside from that, the more democratic structure of the parameter set $\{C_{qq}, C_{gg}, C_{gq}\}$ implies that the high-level and low-level observables show more similar performance.

This comparison between the high-level and the low-level observables should be taken with a grain of salt. First, we know that pre-processing plays a role for the low-level network input, and one could hope to recover a performance closer to the truth-ordering. Second, the non-Gaussian posterior of $C_{gg}$ from the high-level observables suggests that not all trainings might be as stable as the successful training we show here.

## 4 Hadronization and detector

Obviously, the results from the quark-induced toy shower are not what we can expect from an LHC analysis. Already for the comparison with the LEP measurements we need to include hadronization rather than cutting off the QCD splittings at a fixed scale of 1 GeV. In addition, we know that the LHC detectors cannot compete with the $e^+e^-$ environment, but on the other hand the available number of jets will eventually be much larger. Given the promising results for the toy shower the question is how well the analysis would work in a more realistic environment.

For a more realistic simulation we turn to a modified version of SHERPA 2.2.10 [65] and again generate the process

$$e^+e^- \to q\bar{q}, \qquad \text{with} \quad q = u, d, s, \tag{20}$$

without the weakly decaying heavy quarks. The leptonic initial state plays no role for our jet analysis and allows us to ignore initial state radiation. The parton shower is modified to

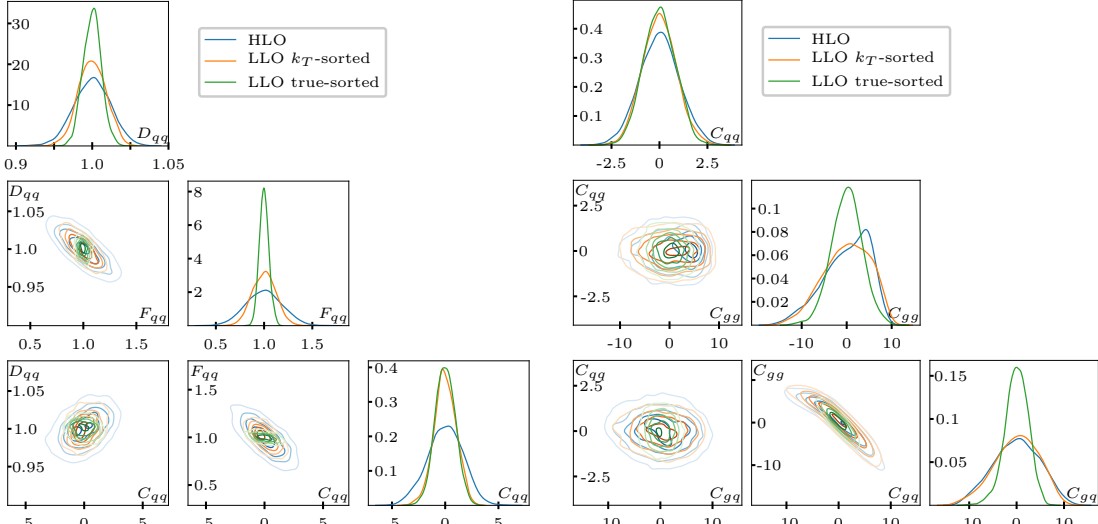

Figure 6: Posterior probabilities from the low-level observables with two sortings and the high-level observables given in Eqs.(18) and (19). In the left panel we assume gluon radiation only, $\{D_{qq}, F_{qq}, C_{qq}\}$, corresponding to Fig. 2. In the right panel we measure the $p_T$-suppressed terms in all QCD splittings, $\{C_{qq}, C_{gg}, C_{gq}\}$, corresponding to Fig. 5.

include our parameterized splitting functions and has a cutoff at 1 GeV. Within SHERPA we still use the modified splitting kernels of Eq.(11) and vary different parameter sets while setting all the others to their SM-values. Unlike for the toy shower we do not remove QCD splittings for the SHERPA case. Without a detector simulation we save the 4-momenta of hadrons, photons and charged leptons. The maximum number of constituents, jets per parameter points etc are identical to our toy shower analysis, and our $k_T$-sorting algorithm is applied to these 4-momenta. In a second step we include LHC detector effects using DELPHES 3.4.2 [80] with the default ATLAS card. Now we save the 4-momenta of all particle flow (PF) objects. The jets are constructed with FASTJET 3.3.4 [81], either processing the hadronization output or the DELPHES output as $R = 1.2$ anti-$k_T$ jets, giving the spectrum

$$p_{T,j} = 20 \text{ GeV} \dots \frac{m_Z}{2} \,. \tag{21}$$

We select only one jet per event for the jet sample. By LHC standards these jets are soft, and since we are testing the structure of QCD splittings, harder jets would include much more information. Because this additional information will at some point be balanced by challenging the calorimeter resolution we stick to this probably over-conservative setup. We also ignore underlying event and pile-up, because standard tools are going to be far from an optimal working point for subjet analyses using low-level observables.

To illustrate the physics behind our proposed measurement, we show the high-level observables from Eq.(18) for the toy shower, after hadronization, and after detector effects in Fig. 7. The bands are defined by a variation $D_{qq} = 0.5. \dots 2$, to illustrate the dependence on the splitting kernels. The number of constituents $n_{\text{PF}}$ generally increases with $D_{qq}$. The toy shower does not generate a very large number of splittings. Hadronization increases the number of constituents significantly, but this effect has nothing to do with QCD splittings. The detector simulation with its resolution and thresholds again leads to a slight decrease. The width of the constituent distribution, $w_{\text{PF}}$, is small for the toy shower, with a peak once the toy shower generates enough splittings. An increase in $D_{qq}$ moves the distribution away from very small

values. Hadronization enhances the peak around $w_{\text{PF}} \approx 0.2$, driven by the hadron decays, and the detector effects have a limited effect because of the explicit $p_T$-weighting. For $p_T D$ a single hard object gives $p_T D = 1$ and adding a soft constituent leads to a downward shift. The small number of QCD splittings leads to a second peak structure around $p_T D \sim 0.7$ for the toy shower, but the entire toy-level distribution has to be taken with a grain of salt. Hadronization then induces the typical shape with a broad maximum below 0.5, again with little impact from the detector effects. Finally, the constituent-constituent correlation $C_{0.2}$ loses all toy-level events at small values when we include hadronization, and the broad feature around $C_{0.2} \sim 0.4$ becomes more narrow and moves to values around 0.6. As a side remark, this variable is particularly effective to distinguish jets from hard quarks and hard gluons, because the two peak structures are relatively well separated with gluons giving larger values of $C_{0.2}$.

The main message from Fig. 7 is that from a QCD point of view the hadronization effects are qualitatively and quantitatively far more important than the detector effects. Therefore, we split our study into two parts. First, we shift from the toy shower to the full SHERPA shower [65], including hadronization. Next, we add detector effects using DELPHES [80] with the default ATLAS card. Unlike for the toy shower, we now vary the parameters for gluon

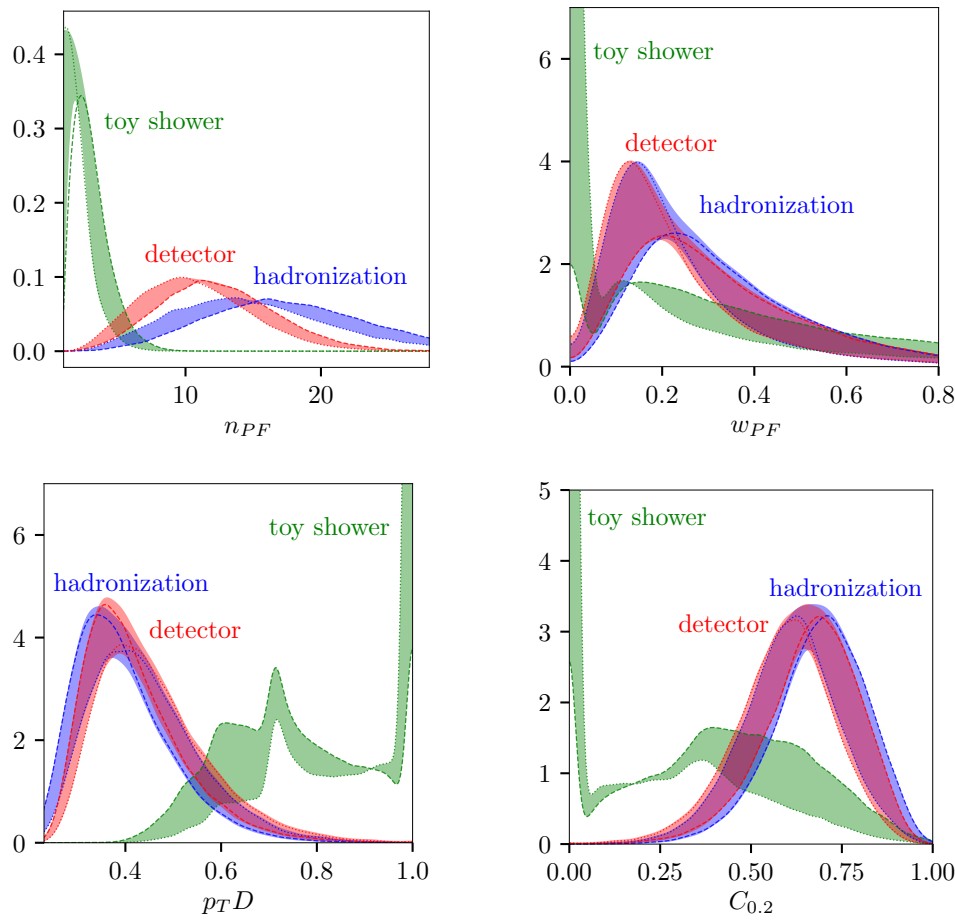

Figure 7: High-level observables $n_{PF}$, $w_{PF}$, $p_T D$, and $C_{0.2}$ for 100k jets. We show results for the toy shower, the SHERPA shower with hadronization, and including detector effects with DELPHES. Bands show the variation of $D_{qq} = 0.5 \ldots 2$ (dotted and dashed).

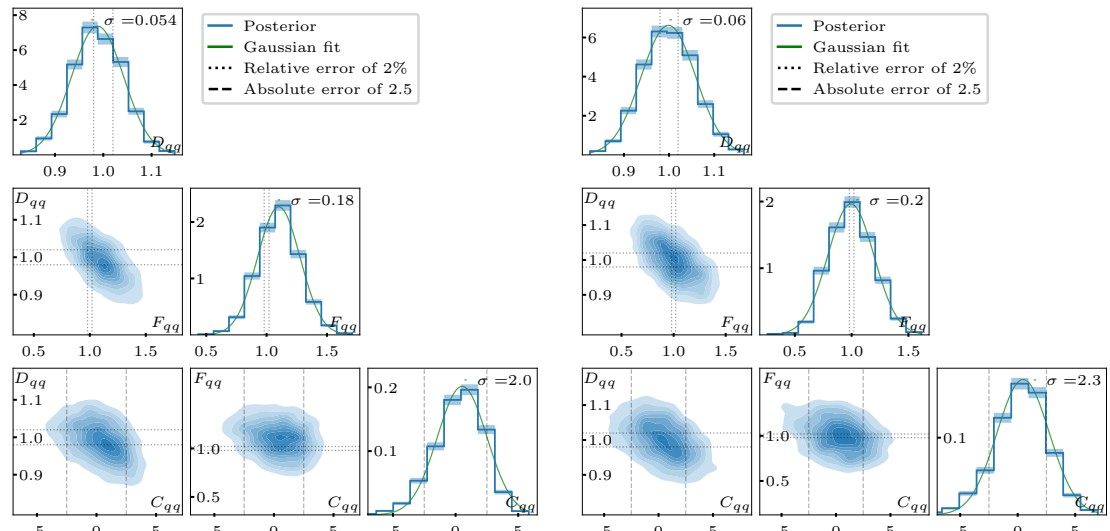

Figure 8: Posterior probabilities for the SHERPA shower, varying the gluon radiation parameters only, $\{D_{qq}, F_{qq}, C_{qq}\}$. We assume SM-like jets and show results without DELPHES detector simulation (left) and including detector effects (right).

radiation,

$$\{D_{qq}, F_{qq}, C_{qq}\} \qquad \text{(gluon radiation varied)}, \qquad (22)$$

while keeping the other splittings fixed to their SM-values. The results are shown in Fig. 8. We apply our $k_T$-sorting throughout this section, to ensure that there is no information backdoor. Compared to the toy-shower in Fig. 2 all error bars for the SHERPA shower are increased by a factor two to four. Intriguingly, the detector resolution adds very little to the uncertainties in the case of our relatively soft jets.

Throughout our analysis we have always assumed that testing our network on SM-like jets is representative of the whole parameter range the model is trained on. For this three-parameter case with a variable quark-gluon splitting we also evaluate 1000 measurements over the whole range covered by the prior. For each parameter point we generate $M = 10^4$ showers, sample 2000 points from the latent space to the measurement, and identify the actual measurement with the average over these 2000 measurements. In Fig. 9 we correlate the true values with the measured values and find that they track each other without a bias, but with a spread corresponding to the known error bars.

Next, we allow for all QCD splittings included in the SHERPA shower and measure the leading soft-collinear contributions, corresponding to measuring $C_F$ and $C_A$ from a sample of quark-induced showers. The combined measurement of the varied parameters

$$\{D_{qq}, D_{gg}\} \qquad \text{(soft-collinear varied)} \qquad (23)$$

is shown in Fig. 10 and can be directly compared to the toy shower results from Fig. 4. Here we see a significant degradation of the SHERPA measurements, especially in $D_{gg}$. This is at least partly due to the correlation between the two extracted model parameters which we do not observe for the toy setup. This correlation is, if anything, slightly enhanced by the detector effects, but as before the effect of the hadronization clearly dominates. For an actual LHC measurement the correlation could be easily removed by combining a quark-dominated and a gluon-dominated jet sample. This is why we also report the measurement for one free model

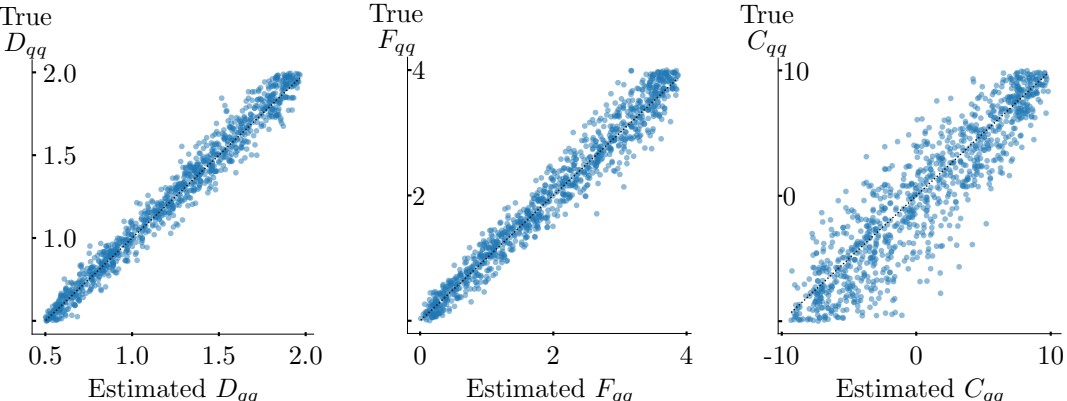

Figure 9: Posterior means vs true values of $D_{qq}$ (left), $F_{qq}$ (middle) and $C_{qq}$ (right) for a test data set with 1000 parameters sets drawn from the prior with 10000 events each calculated including detector simulations in analogy to Fig. 8.

parameter at a time in Tab. 2, indicating that at the LHC we might be able to measure the leading contributions to the splitting kernels in Eq.(11) to the few per-cent level.

Our final hypothesis is that we know the leading and constant terms in all QCD splitting functions, but want to measure possible deviations. This corresponds to varying and then extracting the explicitly $p_T$-suppressed parameters

$$\{C_{qq}, C_{qg}, C_{gg}\} \qquad \text{(rest terms varied)} \qquad (24)$$

from the full SHERPA shower. Unlike for the other parameters, the rest terms are defined around zero, with an explicit $p_T$-suppression. This way we can argue that we do not expect any of the $C_{ij}$ to be significantly larger than one. If that should be the case, our expansion is not correct and the perturbative QCD description of the respective jet sample has a problem. Just looking at $\sigma(C_{qq})$, where the error is roughly a factor two larger than for the toy shower in Fig. 5, this task is clearly in reach, even for our conservative setup. The two other rest terms are essentially invisible. Some of this can likely be cured by combining a quark-dominated and a gluon-dominated sample, where the latter will provide a measurement of $C_{gg}$.

In Tab. 2 we collect all numerical results from this paper. We test three hypotheses, (i) all terms in the gluon radiation off a quark, (ii) the leading terms of the quark and gluon splitting, corresponding for instance to the color Casimirs $C_F$ and $C_A$, and (iii) the $p_T$-suppressed rest terms of the three splitting kernels. The results for the toy shower indicate that the information on the QCD splitting kernels is indeed included in the low-level observables. Obviously, it is easier to extract the leading, regularized pole terms than the finite terms. The rest terms are expected to be zero for the perturbative prediction, so we expect them to be at most of order one. One of the interesting results from the toy shower is that the multi-dimensional posteriors show hardly any correlations, except for the two rest terms $C_{gq}$ and $C_{gg}$. This correlation could be cured by adding gluon-dominated showers in our analysis. Moving to the SHERPA shower we first notice that hadronization has a much more degrading effect than detector effects. While it is still possible to determine the splitting function for gluon emission off a quark and the regularized soft-collinear divergences, we do not have enough sensitivity to constrain all three rest terms. However, $C_{qq}$ is within reach, and $C_{gg}$ should be testable if we include a gluon-dominated jet sample.

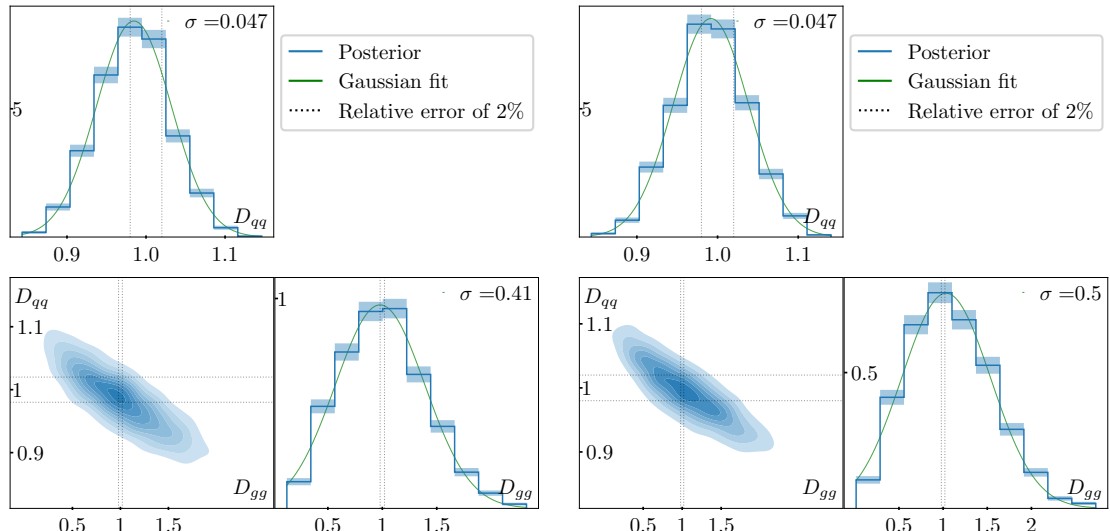

Figure 10: Posterior probabilities for the SHERPA shower, soft-collinear leading terms for all QCD splittings, $\{D_{qq}, D_{gg}\}$. We assume SM-like jets and show results without DELPHES detector simulation (left) and including detector effects (right).

# 5 Outlook

The gold standard of LHC physics is our ability to understand all aspects of the recorded events in terms of fundamental physics. Parton showers, or parton splittings, are part of every LHC analysis. In spite of an active subjet physics program and in spite of significant theoretical progress, we do not have a systematic set of measurements of their simple underlying QCD predictions, even though similar analyses based on jet and event shapes do exist from LEP.

In this paper we have proposed a systematic approach to measuring QCD splittings, including an appropriate technique based on modern machine learning. To define a viable and consistent theory hypothesis, we have parameterized the known splitting kernels into the leading, logarithmically enhanced term, the finite term known in perturbative QCD, and a rest term.

Table 2: Error on the extracted QCD splitting kernels from 10k events in the different setups: gluon radiation only, soft-collinear leading contributions, and $p_T$-suppressed rest terms. The truth-sorting includes an information backdoor through the ordering of the inputs. The asterisk denotes a non-Gaussian posterior. The error in parentheses assumes one variable splitting parameter at a time.

| Setup & Parameter | | Toy shower | | | SHERPA | |
|---|---|---|---|---|---|---|
| | | Truth-sorted | $k_T$-sorted | HLO | Hadronized | Detector-level |
| $\{D_{qq}, F_{qq}, C_{qq}\}$ | $\sigma(D_{qq})$ | 0.012 (0.013) | 0.019 (0.013) | 0.024 (0.015) | 0.054 (0.025) | 0.060 (0.03) |
| | $\sigma(F_{qq})$ | 0.05 (0.04) | 0.16 (0.07) | 0.19 (0.08) | 0.18 (0.09) | 0.20 (0.1) |
| | $\sigma(C_{qq})$ | 0.97 (0.8) | 1.04 (0.8) | 1.7 (1.0) | 2.0 (1.2) | 2.3 (1.4) |
| $\{D_{qq}, D_{gg}\}$ | $\sigma(D_{qq})$ | 0.013 (0.013) | 0.013 (0.013) | 0.013 (0.013) | 0.047 (0.025) | 0.047 (0.025) |
| | $\sigma(D_{gg})$ | 0.034 (0.034) | 0.033 (0.033) | 0.035 (0.035) | 0.41 (0.23) | 0.50 (0.25) |
| $\{C_{qq}, C_{gg}, C_{gq}\}$ | $\sigma(C_{qq})$ | 0.86 (0.8) | 0.90 (0.8) | 1.0 (1.0) | 1.5 (1.0) | 1.4 (0.9) |
| | $\sigma(C_{gg})$ | 3.4 (1.4) | 5.6 (1.7) | 5.4* (1.7) | * * | * * |
| | $\sigma(C_{gq})$ | 2.7 (1.1) | 4.9 (1.4) | 5.2* (1.4) | * * | * * |

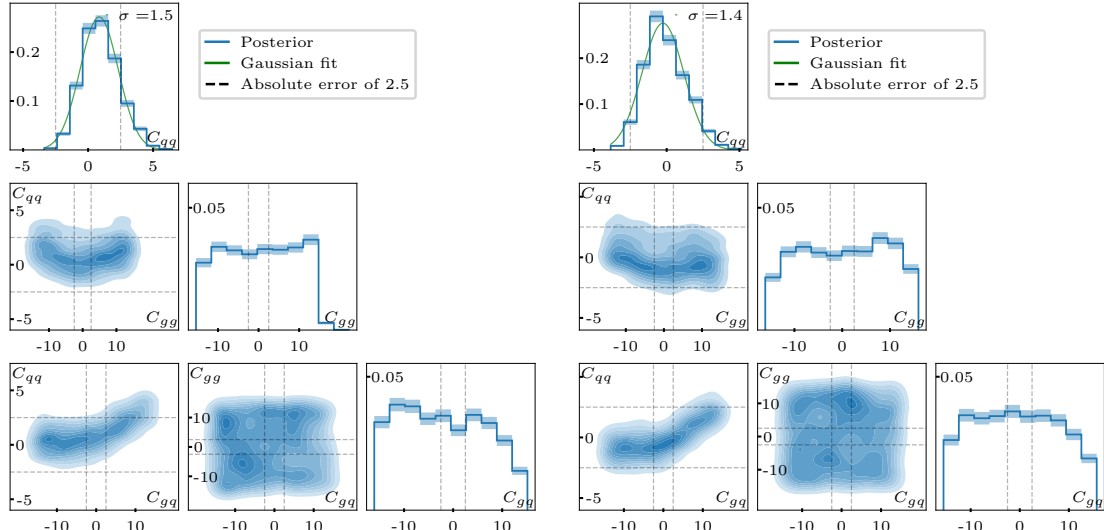

Figure 11: Posterior probabilities for the SHERPA shower, $p_T$-suppressed rest terms for all QCD splittings, $\{C_{qq}, C_{gg}, C_{gq}\}$. We assume SM-like jets and show results without DELPHES detector simulation (left) and including detector effects (right).

Traditionally, the leading term could be identified with the measurement of QCD color factors. The finite term reflects the simple description of parton splittings as a Markov process, while the rest term would allow us to parameterize for instance quantum effects or higher splittings. Expanding the theory hypothesis accordingly would be a natural step in refining any simulation-based analysis.

We have then shown that for a toy shower, modelled after SHERPA, we can measure all these contributions from low-level observables of a jet sample. For a realistic version we saw that hadronization has the biggest effect on our measurement, bigger than the expected detector effects for relatively soft jets. The challenge will be to extract the rest terms beyond the standard QCD predictions, to test the quality of the perturbative QCD prediction.

Our analysis method is based on machine learning, specifically an invertible network conditioned on a small summary network. After training, we can use the invertible network to sample the model parameter space and construct a posterior probability based on a sets of jets. While we study SM-like jets throughout our analysis, the network produces the correct posterior for all jets covered by the original parameter scan.

Our analysis is not meant to be the final word on ML-measurements of fundamental QCD properties from LHC jets. Natural next steps, aside from testing our methodology on actual data would be a second, gluon-initiated jet sample and an additional harder jet sample. While the former will get rid of the remaining correlations in the model parameters, the latter should allow us to optimize the interplay of the energy range covered by the shower and the calorimeter resolution. Our current setup is also not efficient in analyzing millions of jets, because unlike standard likelihood methods it does not scale well with additional data. This is the downside of directly extracting the posterior distribution.

# Acknowledgements

We would like to thank Uli Uwer for fun discussions in general, on the superiority of LHCb, and on the corresponding LEP measurements. The research of AB and TP is supported by the

Deutsche Forschungsgemeinschaft (DFG, German Research Foundation) under grant 3960217 62 – TRR 257 *Particle Physics Phenomenology after the Higgs Discovery*. The authors acknowledge support by the state of Baden-Württemberg through bwHPC and the German Research Foundation (DFG) through grant no INST 39/963-1 FUGG (bwForCluster NEMO). This research was supported by the Fermi National Accelerator Laboratory (Fermilab), a U.S. Department of Energy, Office of Science, HEP User Facility. Fermilab is managed by Fermi Research Alliance, LLC (FRA), acting under Contract No. DE–AC02–07CH11359.

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
