# Peer review of "Measuring QCD Splittings with Invertible Networks"

_SciPost Physics, doi:SciPost Phys. 10, 126 (2021)_

## Round 1 · Referee Report · Anonymous · 2021-2-17

Report

The authors present a machine learning approach to study QCD splittings with conditional and invertible neural networks. I consider their work a useful extension of the growing machine learning literature in high-energy physics. However, there are several points that the authors should address before I can recommend their work for publication.

- Fundamental parameters in QCD (alpha_s or the quadratic SU(3) Casimir invariants CA/CF, as mentioned in the introduction) are typically extracted from cross sections which can be calculated perturbatively to NNLO+NNLL or better. Using a parton shower, where we do not have the same level of perturbative accuracy, would make such precision extractions difficult. Therefore, it seems that high-level but well-controlled observables are a better choice instead of trying to make use of all the low-level information that we get from collider experiments. The discussion in the introduction about precision extractions of CA/CF from LEP data and the relation to the proposed framework of the authors should be further clarified.

- From the introduction, it appears that it is generally difficult to understand the main purpose of the paper. Is it precision extractions, as mentioned above, or the tuning of parameters of parton showers (here, the prefactors introduced in the parametrized splitting functions)? Moreover, it would be helpful for the reader if the authors can comment in more detail on the relationship of their work to the available literature.

- Equation 14 and the sentence before equation 14, “all sub-leading jets are ignored”: Are the authors referring to reconstructed jets or partons from the shower? If these are reconstructed jets, the algorithm and jet radius should be specified. The same question appears in other parts of the paper.

- In the outlook, the authors write that the decomposition of the splitting function in equation 11 corresponds to ``logarithmically enhanced, finite and rest’’ terms. This should be stated more clearly earlier in the paper. How exactly is the functional form of the parametrized splitting functions (in terms of y, z) obtained, especially the term that vanishes in QCD (~C_ij)? I suppose that the decomposition of the logarithmically enhanced terms and the finite terms is not unique and partial fractioning can be used to reshuffle the terms?

- Lastly, I have a question about the k_T sorting which is mentioned several times throughout the paper. Are the authors referring to the ordering from the shower (which is not accessible in experiments like the ``truth sorting’’) or with the help of a clustering/declustering procedure of a jet algorithm (which is accessible experimentally but not directly related to the actual shower history)? Do the considered partons correspond to particles inside a reconstructed jet or to the entire event?

  • validity: -
  • significance: -
  • originality: -
  • clarity: -
  • formatting: -
  • grammar: -

Author:  Tilman Plehn  on 2021-03-19  [id 1318]

(in reply to Report 1 on 2021-02-17)

-> First of all, we would like to thank the referee for their helpful comments. We attach the latexdiff output reflecting our changes.

  • Fundamental parameters in QCD (alpha_s or the quadratic SU(3) Casimir invariants CA/CF, as mentioned in the introduction) are typically extracted from cross sections which can be calculated perturbatively to NNLO+NNLL or better. Using a parton shower, where we do not have the same level of perturbative accuracy, would make such precision extractions difficult. Therefore, it seems that high-level but well-controlled observables are a better choice instead of trying to make use of all the low-level information that we get from collider experiments. The discussion in the introduction about precision extractions of CA/CF from LEP data and the relation to the proposed framework of the authors should be further clarified.

-> We agree with this comment and clarify some of the aspects in the introduction. In essence, our method is simulation-based in the sense that the underlying hypothesis or level of accuracy will be determined by the underlying simulation, which indeed has to be consistent with the data set we are looking at. We also added a brief comment after Eqs.(11,12).

  • From the introduction, it appears that it is generally difficult to understand the main purpose of the paper. Is it precision extractions, as mentioned above, or the tuning of parameters of parton showers (here, the prefactors introduced in the parametrized splitting functions)? Moreover, it would be helpful for the reader if the authors can comment in more detail on the relationship of their work to the available literature.

-> As mentioned above, we have re-organized the introduction to reflect the motivation of our paper better. We also modified parts of Sec.3 and the outlook accordingly.

  • Equation 14 and the sentence before equation 14, “all sub-leading jets are ignored”: Are the authors referring to reconstructed jets or partons from the shower? If these are reconstructed jets, the algorithm and jet radius should be specified. The same question appears in other parts of the paper.

-> In the toy example a jet reconstruction is not needed as only a single parton shower is generated. We phrased this more precisely around Eq.(14). This should be clear for the rest of section 3.

  • In the outlook, the authors write that the decomposition of the splitting function in equation 11 corresponds to logarithmically enhanced, finite and rest’’ terms. This should be stated more clearly earlier in the paper. How exactly is the functional form of the parametrized splitting functions (in terms of y, z) obtained, especially the term that vanishes in QCD (~C_ij)? I suppose that the decomposition of the logarithmically enhanced terms and the finite terms is not unique and partial fractioning can be used to reshuffle the terms?

-> We have commented on this question in more detail, indeed the ansatz of Eq.(11) is not final, but in the current form we believe it should be unique.

  • Lastly, I have a question about the k_T sorting which is mentioned several times throughout the paper. Are the authors referring to the ordering from the shower (which is not accessible in experiments like the truth sorting’’) or with the help of a clustering/declustering procedure of a jet algorithm (which is accessible experimentally but not directly related to the actual shower history)? Do the considered partons correspond to particles inside a reconstructed jet or to the entire event?

-> We improved the description of the k_T sorting and made it more visible on p.8.

Attachment:

diff.pdf

Anonymous on 2021-03-31  [id 1338]

(in reply to Tilman Plehn on 2021-03-19 [id 1318])

I would like to thank the authors for the clarifications and I can recommend their paper for publication.

---

## Editorial Decision

published